# Evolution of Policy Concerning the Readjustment of Inefficient Urban Land Use in China Based on a Content Analysis Method

**Yang Bai** [1] , **Wei Zhou** [1,2,3,*], **Yanjun Guan** [1], **Xue Li** [1], **Baohua Huang** [4], **Fengchun Lei** [4], **Hong Yang** [4] **and Wenmin Huo** [5]

1   School of Land Science and Technology, China University of Geosciences Beijing, Beijing 100083, China; baiyang_bes@cugb.edu.cn (Y.B.); guanyj@cugb.edu.cn (Y.G.); lixue@cugb.edu.cn (X.L.)
2   Key Laboratory of Land Consolidation and Rehabilitation, Ministry of Natural Resources, Beijing 100035, China
3   Technology Innovation Center for Ecological Restoration in Mining Areas, Ministry of Natural Resources, Beijing 100083, China
4   Land Consolidation and Rehabilitation Center, Ministry of Natural Resources, Beijing 100035, China; huangbaohua@lcrc.org.cn (B.H.); leifengchun@lcrc.org.cn (F.L.); yanghong@lcrc.org.cn (H.Y.)
5   Chinese Academy of Natural Resource Economics, Beijing 101149, China; wmhuo@calre.org.cn
*   Correspondence: zhouw@cugb.edu.cn; Tel.: +86-010-8232-1867

**Abstract:** In a 21st century context characterized by the inefficient use of urban construction land, to optimize land use structure and transform resource utilization methods, it is necessary for the Chinese government to improve land use efficiency. Promoting the readjustment of inefficient urban land use has become essential. The purpose of this paper is to sort through the policies addressing the readjustment of inefficient urban land use in China, identify the promulgation date, longitudinal hierarchy, and horizontal composition of the relevant policies, and summarize the evolutionary trend of such policies. This study aims to provide a reference for the adjustment and improvement of relevant policy systems in China. The content analysis method was used in this study, and NVivo 12 software was used to compare and verify the analysis process. The results show the following: (1) A large number of relevant policies have been promulgated, with a total of 12 items from 1988 to 2012, increasing to six items annually on average from 2013 to 2019. The Communist Party of China Central Committee, the National People's Congress of the People's Republic, the State Council and relevant ministries promulgated 13.46%, 11.54%, 28.85%, and 46.15%, respectively, of the relevant policies. (2) The existing policies are mostly issued in the form of notices, opinions, management regulations, etc.; therefore, their level of effectiveness is low, and the role of norms is weak. (3) Finally, the policy content experienced stages of initial exploration, basic establishment, and continuous improvement. It also covered aspects such as Standard Control, Market Configuration, Overall Planning, Incentive Mechanisms, Income Distribution, and Supervision Evaluation. The system for policies addressing readjustment of inefficient urban land use has basically been perfected, although post approval supervision still needs to be strengthened. Therefore, the process of readjusting inefficient urban land use should be optimized, new modes of readjustment should be constantly explored, and inefficient land use should be addressed across the entire territory.

**Keywords:** land use; urban land; land readjustment; content analysis method; policy evolution

## 1. Introduction

Since China's reform and opening up, land use patterns have been based on incremental expansion mainly aimed towards meeting the needs of rapid economic and social development; the demand for urban land is high, and land resources are becoming an increasingly prominent bottleneck [1–5]. The urbanization rate in 1990 was 26.41%, and the built-up area per capita was at its lowest point. Since then, cities have entered a rapid urbanization period. The urbanization rate and built-up area per capita have increased rapidly and are expected to peak in approximately 2020. Specifically, the built-up area per capita in 2014 was 127.02 square meters per person [6,7], showing that with the steady advancement of Chinese urbanization, the previous expansion-type development has proven to be unsustainable, the development mode of incremental planning based on spatial expansion has changed, and urban development planning has gradually shifted from incremental expansion to stock exploration [8–11]. The development model of inventory planning takes urban renewal as its driver, starting by improving public facilities, optimizing the structure of industrial land, accelerating urban transformation, and upgrading infrastructure [12]. It aims to renovate the inefficient stock of construction land in specific built-up areas of the city. This means that urban development is gradually transforming and upgrading from urban sprawl to the readjustment of inefficient urban land use in China [13]. Given the current status of economic and social sustainable development in China, although the planning of urban land still tends toward the expansion of the existing land stock, a higher level of efficiency is required rather than speed. Intensive, scientific, and balanced land use should become the basic criteria in pursuit of efficiency goals in the development model of inventory planning [14]. Land use efficiency means that when people use land resources as factors for production, they continuously improve the level of production technology and rationally allocate the input ratio of each production factor to increase the use level of land resources and maximize the output of land resources [15,16]. Inefficient urban land refers to urban construction land that has been identified as construction land by the second national land survey. This land has clear ownership, with no controversies or legal disputes, but also has a scattered layout, extensive and unreasonable use, and dilapidated construction [17]. Combining the Ministry of Natural Resources survey results on the readjustment of inefficient urban land use and relevant data from the National Bureau of Statistics, the area of inefficient urban land accounted for 9.85% of the urban built-up area in 2018 [18], and land use efficiency was lower. To achieve the goals of maintaining cultivated land, promoting growth, optimizing the land use structure, and transforming resource utilization methods in China, it is necessary to improve land use efficiency [19–23]. Therefore, promoting the readjustment of inefficient urban land use has become a necessary next step.

Due to differences in development time and focus, some countries refer to the readjustment of inefficient urban land use as "urban renewal" [24–34]. Urban renewal mainly arises in developed Western countries [35–40]. The urban renewal development process of European and American countries represented by the United States, the United Kingdom, and France is mainly divided into three stages: (1) From the 1940s to the 1970s, demolition and reconstruction gave way to comprehensive renovation and renewal, and finally to small-scale, organic renewal in phases [41–43]. For example, in the mid-1960s, the Model City Plan of the United States formulated a comprehensive solution to poverty in several specific areas of large cities. Most government subsidies were used to improve education, medical services, employment, and public safety in low-income communities in urban renewal areas, and the remaining subsidies were used to improve infrastructure and living conditions [44]. (2) In the 1980s, urban renewal changed from being led by the government to being led by the market and then to involving multiple parties [45]. For example, the United Kingdom introduced the Planning and Lands Act in 1980, which allows the establishment of urban development zones and enterprise zones and encourages joint-stock companies that cooperate to participate in urban renewal, thereby activating depressed areas of the inner city [46]. (3) Since the 1990s, urban renewal has changed from focusing on the physical environment to being people oriented and promoting sustainable development [47,48]. For example, France promulgated the Social Solidarity and Urban

Renewal Law in 2000, which established urban renewal as a new urban development model promoting the economical use of space and energy, the rejuvenation of declining urban areas, and the enhancement of social characteristics. Due to differences in socioeconomic environments, urbanization processes, social participation, and coordination mechanisms, the readjustment of inefficient urban land use in coastal areas, such as Guangdong, Zhejiang, Jiangsu, Shanghai, and Hong Kong in China, has certain differences from Western urban renewal. For example, Western urban renewal after the Second World War was based on the rise, development, and maturity of cities. As a whole, it was a process of urban self-healing and improvement. The readjustment of inefficient urban land use in China's coastal areas occurs against the background of increasing urbanization, and the frequent criticisms of this process include that it involves insufficient social participation, weak coordination mechanisms, and increasing readjustment costs. As the development path of Western urban renewal has passed, the readjustment of inefficient urban land use has developed slowly in China, but it has entered a new stage of organic renewal [49,50]. In the new stage of development, large-scale expansion has been transformed, with a focus on the high-quality exploration of land use potential and social development inheriting particular historical and cultural contexts [51–56]. For example, the 798 Art District, located in the Chaoyang district of Beijing, used to be the site of the former state-owned 798 factory and other old factories for the electronics industry. The local government mainly carried out the renovation of the old industrial zone by optimizing the spatial environment of the art zone. Ultimately, the 798 Art District, as an organic combination of contemporary art, architectural space, cultural industry, historical context, and urban living, has become a new landmark of urban culture in Beijing.

However, within the reviewed literature, researchers have observed that most Chinese scholars only focus on policies for the readjustment of inefficient urban land use in pilot cities. Considering existing research, this paper aims to address the following questions: (1) How can the policies of readjustment of inefficient urban land use be analyzed from the national level by using the content analysis method? (2) How can we analyze the entire development process of policy for the readjustment of inefficient urban land use by using the content analysis method [57]?

Overall, studies addressing policy evolution for the readjustment of inefficient urban land use based on the national level play a significant role in the adjustment and improvement of related policies. Therefore, the objectives of this study are as follows: (1) Sort the policies on the readjustment of inefficient urban land use introduced at the national level and use the content analysis method to construct a theoretical framework including policy promulgation date, longitudinal hierarchy, and horizontal composition; (2) Identify the analytical categories and carry out reliability and validity assessments to overcome the subjectivity and uncertainty of qualitative research; (3) Encode the analytical categories and quantitatively analyze the status quo and evolution of policy content; and (4) Use NVivo 12 software to compare and verify the rationality of the analysis process from content analysis methods, and that aims to provide reference for the adjustment and improvement of relevant policy system in China. The innovations of this paper are as follows: (1) We used the content analysis method. Because qualitative research has certain levels of subjectivity and uncertainty, we can use content analysis to convert qualitative analysis into quantitative research, thereby overcoming the disadvantages of qualitative research. (2) Few scholars have systematically analyzed the evolution of policy on the readjustment of inefficient urban land use in China from the national level, and we have filled this gap.

Considering the research questions and objectives, this paper uses the content analysis method to analyze the evolution of policies regarding readjustment of inefficient urban land use in China. This paper is organized into four parts. The next section will focus on the research method and analysis processes of this paper. Section 3 presents analyses of the evolution of policy contents and the results in this paper, including reliability and validity assessments and analyses of changes in policy volume and longitudinal and horizontal policy structures, as well as the validation of NVivo 12 software. This paper ends with a conclusion and discussion in Section 4.

## 2. Research Method and Analysis Processes

### 2.1. Content Analysis Method

The content analysis method was originally applied in the fields of journalism and communication and can be applied to the systematic research of any piece of literature or recorded communication; thus, it is widely used in the social sciences [58]. Through content analysis, documents expressed in linguistic representations rather than quantities are converted into quantities of data by identifying the key information and main features of the target text and describing the analysis results numerically. Based on analysis of the "quantity" of literature content, characteristics that reflect certain essential aspects of the content can be identified and made easy to count, thus overcoming the subjectivity and uncertainty of qualitative research [59]. This method is somewhat subjective in the division of time periods. To reduce the subjective bias of the judges and the resulting influence on the research results, it is necessary to identify the categories and coding rules in the theoretical analysis framework through reliability testing to ensure the scientific rigor of the analysis process and the results. Therefore, it is necessary to select different judges to compare the coding results of the same classification sample, and the consistency of the two must be greater than 80%.

### 2.2. Data Source and Sample Selection

To ensure the orderly readjustment of inefficient urban land use in China, the Communist Party of China (CPC) Central Committee, the National People's Congress (NPC) of the People's Republic, as well as the State Council and its relevant departments have issued a series of rules and regulations and made clear provisions on the overall requirements, incentive mechanisms, and safeguard measures. In this paper, the relevant policies since the reform and opening up are selected as the research data, and the policy documents issued by the abovementioned departmental agencies are used as objective evidence for analyzing the evolution of policies. Through the direct search of official websites, publicly promulgated policies addressing the readjustment of inefficient urban land use are obtained by backtracking and searching for the relevant content in the literature materials and policy texts. To ensure the accuracy and representativeness of the text, the main content or the part of the content that is closely related to the readjustment of inefficient urban land use is selected, and the text is determined to be a legislative document or a normative document, such as a binding regulation, opinion, measure, or notice [60]. In addition, the technical points that have been compiled around the planning of the readjustment of inefficient urban land use are included in the analysis. Ultimately, 52 samples of effective policies at the national level are identified (Table 1).

**Table 1.** Policy texts relating to the evolution of the readjustment of inefficient urban land use in China.

| Form | Number | Name of the Policy Text | Brief Summary and Qualitative Judgment of Policy | Date |
|---|---|---|---|---|
| Guidelines of CPC | 1 | Decision of the Communist Party of China Central Committee (CCCPC) on Some Major Issues Concerning Comprehensively Deeping the Reform | Establish space planning system and improve the intensive use of energy, water, and land. | 2013 |
| | 2 | Recommendations of the CCCPC on the 13th Five-Year Plan for Economic and Social Development | Adhere to the strictest land-saving system, adjust the structure of construction land, and promote the readjustment of inefficient urban land use. | 2015 |
| | 3 | Several Opinions of the CCCPC on the Comprehensive Revitalization of Old Industrial Bases in Northeast China | The central government has continued to increase investment for shantytown reconstruction and promote urban renewal. | 2016 |
| | 4 | Opinions of the CCCPC on Strengthening the Protection of Cultivated Land and Improving the Balance of Compensation | Revitalize the stock of construction land and optimize the layout of construction land. | 2017 |
| | 5 | Guiding Opinions of the CCCPC on Coordinating the Reform of the Property Right System of Natural Resource Assets | Play a decisive role in allocating resources in the market by improving the price formation mechanism. | 2019 |
| | 6 | Several Opinions of the CCCPC on Establishing a Territorial Planning System and Supervising Implementation | Optimize the structure and layout of territorial space. | 2019 |
| Related Laws | 7 | Land Administration Law of the People's Republic of China (PRC) | Public facilities and other construction land should be rationally distributed, comprehensively developed, and ancillary. | 2004 |
| | 8 | Urban-Rural Planning Law of the PRC | Reconstruction of old cities should follow the principle of reasonable layout, and it should protect historical and cultural heritage. | 2007 |
| Administrative Regulations | 9 | Tentative Regulations of Urban Land Use Tax of the PRC | Advocates the rational use of urban land and the adjustment of land-level income. | 1988 |
| | 10 | Decision of the State Council on Amending the Tentative Regulations of the Urban Land Use Tax of the PRC | Introduced standards for investigation, identification, disposal, and use of idle land. | 2006 |
| | 11 | Measures for the Administration of Pilot Projects Linked to the Increase or Decrease of Urban and Rural Construction Land | Reasonable use of urban land, adjustment of land-level income, and improvement of land use efficiency. | 2008 |
| | 12 | Disposal Methods for Idle Land | Coordinate the pilot work with planning and guide the structural adjustment and layout optimization of urban and rural land. | 2012 |
| | 13 | Guiding Opinions on Promoting Economical and Intensive Land Use | Intensive land use refers to the effect of land conservation through scale guidance, layout optimization, standard control, market allocation, active utilization, and so on. | 2014 |
| | 14 | Management Measures for the Pilot Project of the Reclamation and Utilization of Abandoned Industrial and Mining Wasteland | Reasonably adjust layout of construction land, and the competent land and resources department should regularly check its work. | 2015 |
| | 15 | Measures for the Administration of Annual Plans on the Utilization of Land | Coordinate stocks and newly added construction land to promote active utilization of stocks. | 2016 |
| | 16 | Measures for the Administration of the Overall Plan for Land Utilization | Optimization plan for land use structure, layout, timing arrangements, and saving for intensive land. | 2017 |

**Table 1.** *Cont.*

| Form | Number | Name of the Policy Text | Brief Summary and Qualitative Judgment of Policy | Date |
|---|---|---|---|---|
| Notices and Opinions | 17 | Notice of the State Council on Promoting the Economical and Intensive Use of Land | Construction land should implement market allocation, control land use standards, strengthen supervision, and establish an assessment system. | 2008 |
| | 18 | Guiding Opinions of the State Council on Accelerating the Development of the Producer Services Industry to Promote the Adjustment and Upgrading of the Industrial Structure | Encourage the development of productive service industries through the transformation of inefficient urban land use and adhere to market leadership. | 2014 |
| | 19 | Several Opinions of the State Council on Deepening the Construction of New Urbanization | Accelerate the transformation of urban shantytowns, urban villages, and dilapidated houses and establish an incentive mechanism for the readjustment of inefficient urban land use. | 2016 |
| | 20 | Notice of the State Council on Printing and Distributing the 13th Five-Year Plan to Promote the Equalization of Basic Public Services | Accelerate the transformation of concentrated shantytowns and urban villages. | 2017 |
| | 21 | Notice of the State Council on Several Measures for Utilizing Foreign Capital to Promote High-Quality Economic Development | Allow localities to support plant transformation and internal land consolidation to improve intensive land use. | 2018 |
| | 22 | Opinion of the State Council on Promoting the High-Quality Development of Innovation and Entrepreneurship to Create an Upgraded Version of Double Innovation | Promote the readjustment of inefficient urban land use, optimize the structure of land use, revitalize the stock and idle land for innovation and entrepreneurship. | 2018 |
| | 23 | Guiding Opinions on Promoting the Relocation and Renovation of Old Industrial Areas in Urban Areas | Promote the transformation of shantytowns in the old industrial areas. | 2014 |
| | 24 | Several Opinions of the State Council on Promoting the Transformation, Upgrading and Innovation of National Economic and Technological Development Zones | Optimize the industrial structure and layout with quality improvement and efficiency upgrading as the core. | 2014 |
| | 25 | Guiding Opinions on Piloting the Readjustment of Inefficient Urban Land Use | Ten provinces were identified as pilots for the readjustment of inefficient urban land use in China. | 2013 |
| | 26 | Notice on the 2013 Action Plan for the Printing and Development Insurance Red Line Project | Deploy and carry out pilots readjusting inefficient urban land use to promote urban renewal. | 2013 |
| | 27 | Guiding Opinions on Promoting Land Economical and Intensive Use | Revitalize the stock land for construction, optimize the layout of land use, improve the land use control standards, and so on. | 2014 |
| | 28 | Notice on Further Doing a Good Job in Land Service Guarantee for New Urbanization Construction | Establish an incentive mechanism for the readjustment of inefficient urban land use to promote the transformation of urban villages. | 2016 |
| | 29 | Guiding Opinions on Implementing the Target of Decreasing the Area of Construction Land for the 13th Five-Year Plan | Efforts will be made to revitalize the stock of construction land and promote the market-oriented allocation of land resources. | 2016 |
| | 30 | Guiding Opinions on Deepening the Readjustment of Inefficient Urban Land Use (Trial) | Adhere to government's guidance, planning first, market orientation, guided by the situation, public participation, and equal consultation. | 2016 |
| | 31 | Notice on the Publication of the Recommended Catalogue (First Batch) of Land-Saving Technologies and Modes | Introduce a number of typical cases involving land-saving technologies and land-saving modes. | 2017 |

**Table 1.** *Cont.*

| Form | Number | Name of the Policy Text | Brief Summary and Qualitative Judgment of Policy | Date |
|---|---|---|---|---|
| Notices and Opinions | 32 | Guidance on Strengthening Urban Rehabilitation Work for Ecological Restoration | Make overall use of funds from various sources to speed up the renovation of old houses. | 2017 |
| | 33 | Implementation Opinions on Supporting the Industrial Transformation and Upgrading of Old Industrial Cities and Resource-Based Cities | Support the inclusion of cities where demonstration areas are located in pilots readjusting inefficient urban land use. | 2016 |
| | 34 | Supporting the Construction of the First Batch of Old Industrial Cities and Resource-Based Cities for Industrial Transformation and Upgrading | Establish a sound evaluation and reward incentive mechanism and dynamic adjustment. | 2017 |
| | 35 | Notice on Printing and Distributing the Promotion Plan for the Readjustment of Inefficient Urban Land Use | Establish the mechanism for monitoring and evaluation, strengthen risk assessment, and promote the readjustment of inefficient urban land use. | 2017 |
| | 36 | Notice on Improving the Mechanism of Increasing Deposits for Construction Land | Effectively dispose of idle land and encourage active utilization through legal transfer and cooperative development. | 2018 |
| | 37 | Notice on Further Improving the Retention, Utilization and Renovation of Existing Urban Buildings | Attach great importance to the preservation, use, and renovation of existing buildings in the city. | 2018 |
| | 38 | Bulletin on the Evaluation of the Conservation and Intensive Use of Urban Construction Land | Strictly control the scale of construction land and continuously optimize the structure of land use. | 2018 |
| | 39 | Bulletin on the Promotion of the Readjustment of Inefficient Urban Land Use | Optimize land use structure and urban spatial layout, improve the incentive policy system, and so on. | 2018 |
| | 40 | Notice on the Second Batch of Land-Saving Technologies and Model Case Collection issued by the Ministry of Natural Resources | Collection of typical cases of readjustment of inefficient urban land use and active use of stock construction land. | 2019 |
| Planning and Reporting | 41 | 10th Five-Year Plan for National Economic and Social Development of the PRC (2001–2005) | Promote the adjustment of the old industrial base structure, optimize the industrial structure and regional layout. | 2001 |
| | 42 | 11th Five-Year Plan for National Economic and Social Development of the PRC (2006–2010) | Strictly control the increment, revitalize the stock, and control the scale of conversion of agricultural land to construction land. | 2006 |
| | 43 | 12th Five-Year Plan for National Economic and Social Development of the PRC (2011–2015) | Revitalize the stock of construction land and optimize the industrial layout. | 2011 |
| | 44 | 13th Five-Year Plan for National Economic and Social Development of the PRC (2016–2020) | Promote the comprehensive development and utilization of aboveground and underground three-dimensional space. | 2016 |
| | 45 | Report on the Work of the Government in 2007 | Control increments and revitalize stocks. | 2007 |
| | 46 | Report on the Work of the Government in 2015 | Intensify the transformation of urban shantytowns. | 2015 |
| | 47 | Report on the Work of the Government in 2018 | Promote the renovation of urban villages and old communities. | 2018 |
| | 48 | Outline of the Master Plan for National Land Use (2006–2020) | Actively revitalize stock construction land, and so on. | 2006 |
| | 49 | Plan of National New-Type Urbanization (2014–2020) | Optimize urban structure of spatial and management, and so on. | 2014 |
| | 50 | Plan of National Land Consolidation (2016–2020) | Strengthen overall planning and guidance, and so on. | 2016 |
| | 51 | Outline of Territorial Planning (2016–2030) | Focus on revitalizing the stock of land, and so on. | 2016 |
| Technical Regulation | 52 | Key Points for the Preparation of Inefficient Urban Land Use Redevelopment Planning | Formulate the identification standards for inefficient urban land use and adhere to layout optimization and benefit sharing. | 2017 |

### 2.3. Policy Text Analysis Framework

After the policy texts concerning the readjustment of inefficient urban land use in China were collected and sorted, the 52 policies were reviewed in regards to three aspects: the promulgation date, longitudinal hierarchy, and horizontal composition of the text (Figure 1). This laid the foundation for the later coding of the policy text. The promulgation date was mainly used to analyze the change in the number of policies issued over time. Longitudinal hierarchy and horizontal composition were used to analyze the structure of the policy subjects and policy texts. The longitudinal hierarchy was based mainly on an analysis of the composition of policies issued by the CPC Central Committee, the NPC of the People's Republic, and the State Council and its related departments. In addition, the quantitative structure was analyzed based on the different policy subjects. The horizontal composition perspective was adopted to analyze the quantitative structure of the policy text form [61].

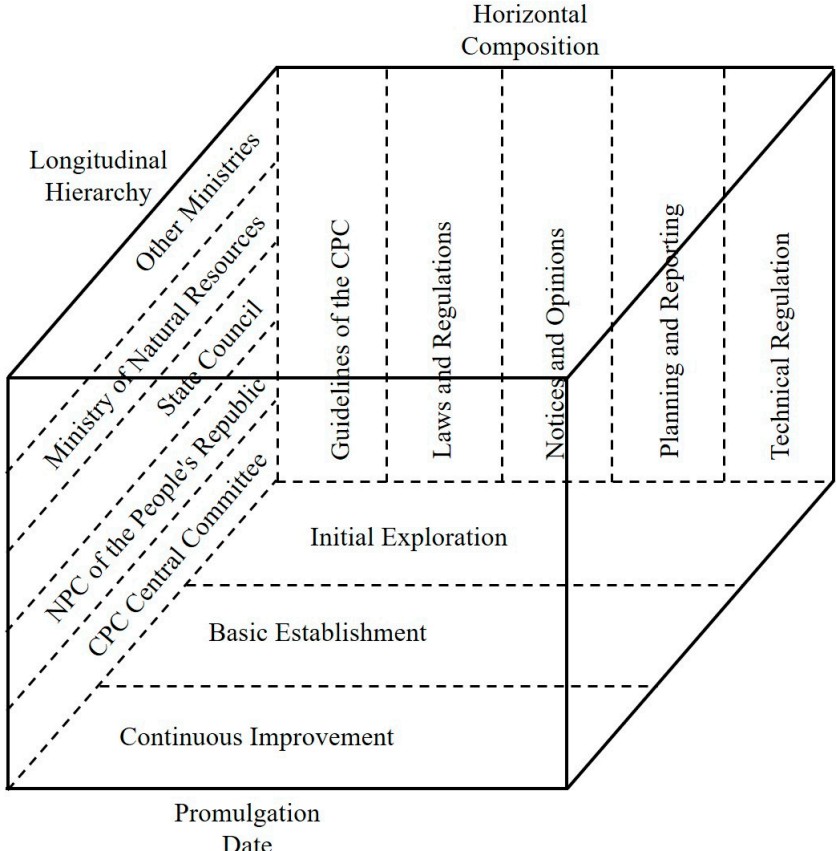

**Figure 1.** Text analysis framework to capture the policy evolution process.

### 2.4. Content Encoding and Frequency Statistics

Based on a comparative analysis of classification systems used in existing research, the analytical categories were identified according to the research objectives and considering three factors: (1) The analytical categories were based on high-frequency keywords appearing in the policy texts addressing the readjustment of inefficient urban land use. (2) Guided by the research aim, the classification of categories needed to closely follow the evolution of policies for the readjustment of inefficient urban land use. (3) Because some policy texts were not clearly attributable to a specific category, the categories were not mutually exclusive. This meant some policy texts belonged to multiple analytical categories simultaneously. (4) Finally, rather than the general classification system in the field of territorial management, the category system was applicable to the policy texts selected in this paper [62]. After combining the above factors, the analytical categories established in this paper were as follows: A-Update and Renovation; B-Revitalization and Utilization; C-Layout Optimization;

D-Overall Planning; E-Standard Control; F-Market Configuration; G-Incentive Mechanism; H-Income Distribution; I-Supervised Evaluation; and J-Legal Liability.

After the analytical categories were established, the 52 policy texts were coded according to the "Text Number-Specific Terms/Chapter". If a certain policy text fell into multiple analytical categories, a serial number was assigned in the order of the extracted policy content, following the format of "Text Number-Specific Terms/Chapter - Serial Number", to finally form the code table of the policy content analysis unit. The frequency of the above analytical categories in the three time periods was counted (Table 2), and a two-dimensional distribution map of the policy texts was drawn (Figure 2). In other words, the distribution of policy text content was established in 10 analytical categories over three time periods.

To compare the frequency statistics of the 10 analytical categories, we used NVivo 12 software, which counts the word frequency for each analytical category. We confirmed that the results obtained through NVivo 12 support our frequency statistics.

| Category code | Before 2013 | 2013-2015 | 2016-2019 |
|---|---|---|---|
| J | | 13-6 | 30-5 |
| I | 17-3 | 13-5; 14-2 | 34-2; 35-3 |
| H | | 25-4; 27-6 | 30-4; 52-3 |
| G | | 25-3; 27-5 | 3-2; 19-2; 28-2; 30-3; 34-1; 35-2; 39-4; 50-5-3 |
| F | 17-2 | 13-4; 18-2; 25-2; 27-4 | 5; 29-2; 30-2 |
| E | 9; 10; 11; 17-1 | 13-3; 27-3; 49-7 | 35-1; 52-2 |
| D | 7-59-2; 12-2; 48-4-4 | 1; 25-1 | 30-1; 39-3; 50-5-2; 51-6 |
| C | 7-59-1; 8-1; 12-1; 41-2-2; 43-3; 48-4-3 | 2; 13-2; 14-1; 24; 27-2; 49-1 | 4-2; 6; 16; 22-2; 31; 38; 39-2; 50-1; 51-1; 52-1 |
| B | 42-6; 43-6; 45; 48-4-2 | 13-1; 27-1 | 4-1; 15; 22-1; 29-1; 36; 40; 44-10; 51-4 |
| A | 8-3; 41-2-1; 48-4-1 | 18-1; 23; 26; 46; 49-5 | 3-1; 19-1; 20; 21; 28-1; 32; 33; 37; 39-1; 47; 50-5-1 |

Time period of promulgation

**Figure 2.** Distribution of policies by form. The capital letters represent different analytical categories, such as A means Update and Renovation, B means Revitalization and Utilization, and so on. The digit in the row of each analytical category represents the number of the policy text within the corresponding analytical category, e.g., the analytical category is the first line number 41-2-1 corresponding to code A, which means that the analytical category is the policy text under Update and Renovation. The number of this policy text is 41, the specific term/Chapter is 2, and the serial number is 1, thus forming the format of "Number-Specific Terms/Chapter-Serial Number".

**Table 2.** The frequency and percentage of policy texts.

| Number | Code | Analysis Category | Before 2013 | | 2013–2015 | | 2016–2019 | | Total | |
|---|---|---|---|---|---|---|---|---|---|---|
| | | | Frequency | Percentage/% | Frequency | Percentage/% | Frequency | Percentage/% | Frequency | Percentage/% |
| 1 | A | Update and Renovation | 3 | 13.64 | 5 | 17.24 | 11 | 21.57 | 19 | 18.63 |
| 2 | B | Revitalization and Utilization | 4 | 18.18 | 2 | 6.90 | 8 | 15.69 | 14 | 13.73 |
| 3 | C | Layout Optimization | 6 | 27.26 | 6 | 20.68 | 10 | 19.61 | 22 | 21.57 |
| 4 | D | Overall Planning | 3 | 13.64 | 2 | 6.90 | 4 | 7.84 | 9 | 8.82 |
| 5 | E | Standard Control | 4 | 18.18 | 3 | 10.34 | 2 | 3.92 | 9 | 8.82 |
| 6 | F | Market Configuration | 1 | 4.55 | 4 | 13.79 | 3 | 5.88 | 8 | 7.85 |
| 7 | G | Incentive Mechanism | 0 | 0.00 | 2 | 6.90 | 8 | 15.69 | 10 | 9.80 |
| 8 | H | Income Distribution | 0 | 0.00 | 2 | 6.90 | 2 | 3.92 | 4 | 3.92 |
| 9 | I | Supervised Evaluation | 1 | 4.55 | 2 | 6.90 | 2 | 3.92 | 5 | 4.90 |
| 10 | J | Legal Liability | 0 | 0.00 | 1 | 3.45 | 1 | 1.96 | 2 | 1.96 |
| | | Total | 22 | 100% | 29 | 100% | 51 | 100% | 102 | 100% |

## 3. Results Analysis

### 3.1. Reliability and Validity Assessment

The reliability of the content analysis method, that is, the degree of consistency between the measurement results, includes the reliability of the defined category and the reliability between the judges. In this paper, three professors and two associate professors acted as judges to determine the analytical categories. They all have major achievements in intensive land use, urban-town-village land layout optimization control, land use and sustainable development, which are subjects close to the research scope of the readjustment of inefficient urban land use, and they were selected as judges in line with scientific and reasonable research purposes. The results show that the evaluation panel fully agrees with the analytical categories established in this paper, and according to the principle for the evaluation of interrater reliability adopted in the relevant literature, the obtained reliability is 0.89, which is within the reasonable interval [0.8,0.9] defined by VINEY [63].

The validity of the content analysis method refers to the extent to which the empirical measurement reflects the true meaning of the concept, including validity of the category and of the analytical framework. In the validity evaluation, five experts were invited to evaluate the degree to which the analytical categories and framework in this paper reflect policy for the readjustment of inefficient urban land use by assigning a value from 0–5 [64]. According to the evaluation results, the validity of the analytical category reached 4.72, and scores for the effectiveness of the analytical framework were 5, 5, 5, 4, and 5, which means that the constructed categories and framework fitted the existing policy system well and can be used for analysis.

### 3.2. Analysis of Changes in Policy Volume

Since the reform and opening up process in China, the number of policies for the readjustment of inefficient urban land use has shown an upward trend (Figure 3). Based on a systematic study of 52 related policies, 1988, 2013, and 2016 represent historical turning points in the readjustment of inefficient urban land use in China. Only 12 policy texts covering the readjustment of inefficient urban land use were issued between 1988 and 2012, most of which were laws, regulations, and planning reports. A total of 12 relevant policies were issued during 2013–2015, accounting for 23% of the total policy texts on the readjustment of inefficient urban land use in China. The number of relevant policies introduced in 2016–2019 increased dramatically, with a total of 28 items and an average of seven items per year, accounting for 54% of the total policy texts for the readjustment of inefficient urban land use in China; most of these are notices. Over time, inefficient urban land use has become an important issue that cannot be ignored. The Chinese government has successively introduced a number of relevant policies for regulation and guidance.

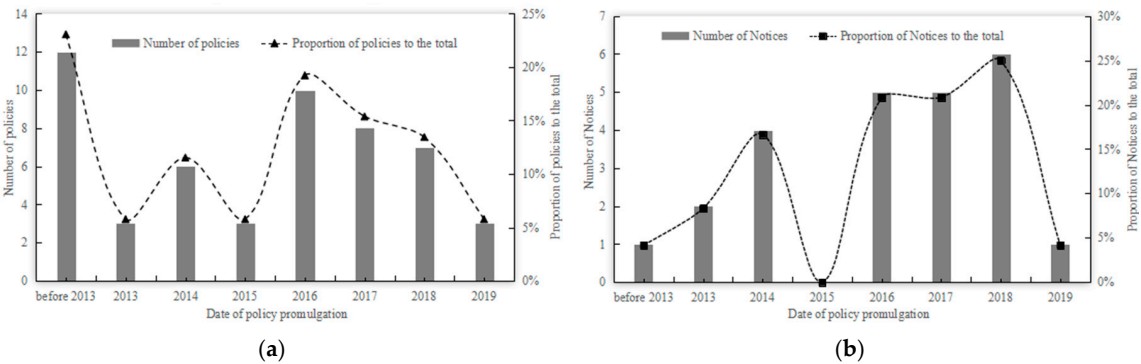

**Figure 3.** Changes in the number and proportion of relevant policies: (**a**): Changes in the number and proportion of total policies; (**b**): Changes in number and proportion of policies in the form of Notices.

### 3.3. Analysis of the Longitudinal and Horizontal Policy Structure

The social network analysis function in NVivo 12 software can focus on connections between various social entities. According to the social network analysis (Figure 4), the NPC of the People's Republic and the State Council are important nodes in the relationship. The CPC Central Committee acts as the top designer in the policy-making process in relation to readjustment of inefficient urban land use. Accordingly, a "center-periphery" structure exists among the multi-sector network in China, which has formed the model of readjustment of inefficient urban land use with the CPC Central Committee as the core, the NPC of the People's Republic and the State Council as the central nodes, and the Ministry of Natural Resources and other Ministries as active responders.

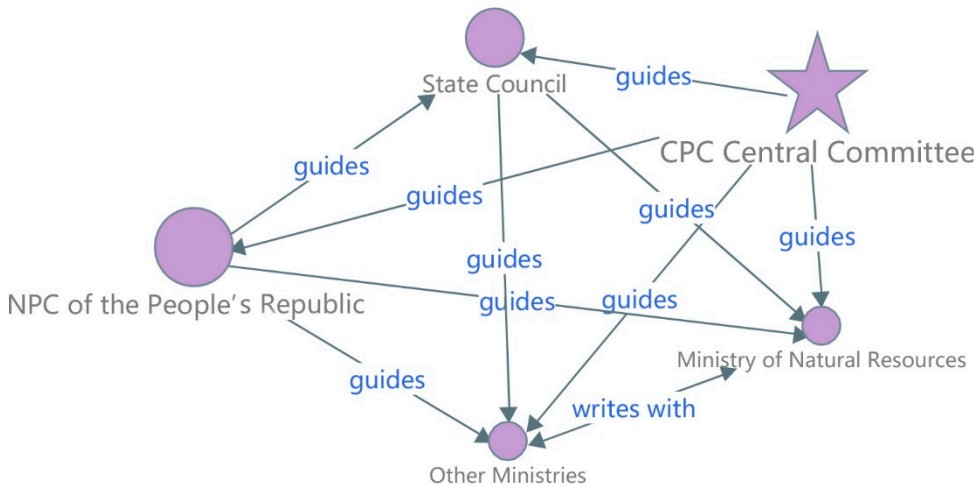

**Figure 4.** Social network analysis of policy-making entities.

Judging from the composition of the policy-issuing institutions (Table 3), state-level policies for the readjustment of inefficient urban land use are mostly issued by the State Council and its relevant ministries, which have issued 75% of the total relevant policies in China. In contrast, although the policies issued by the CPC Central Committee and the NPC of the People's Republic are relatively few in number, accounting for only 13.46% and 11.54% of the total texts, according to Table 4, the relevant policy texts emphasize the purpose, process, and enforcement of the readjustment of inefficient urban land use, which shows that the CPC Central Committee attaches great importance to this policy.

**Table 3.** Composition of policies on the evolution of readjustment of inefficient urban land use.

| Institutional Composition | CPC Central Committee | NPC of the People's Republic | State Council | Relevant Ministers of the State Council | | | Total |
|---|---|---|---|---|---|---|---|
| | | | | Subtotal | Ministry of Natural Resources | Other Ministries | |
| Quantity | 7 | 6 | 15 | 24 | 20 | 4 | 52 |
| Percentage/% | 13.46 | 11.54 | 28.85 | 46.15 | 38.46 | 7.69 | 100 |
| Text Composition | Guidelines of the CPC | Laws and Regulations | Notices and Opinions | Planning and Reporting | | Technical Regulation | Total |
| Quantity | 6 | 10 | 24 | 11 | | 1 | 52 |
| Percentage/% | 11.54 | 19.23 | 46.16 | 21.15 | | 1.92 | 100 |

In addition, among the policies issued by the relevant ministries of the State Council, the Ministry of Natural Resources issued 20 items, and the other four were issued by other ministries of the State Council. Combining the purpose, process, and enforcement of policy-issuing institutions (Table 4), it can be seen that the readjustment of inefficient urban land use has shown a good form of multisectoral co-management in China.

Based on the horizontal compositions of policies (Table 3), policies regarding readjustment of inefficient urban land use are mainly guided by a large number of "notices" and "opinions" under the laws, regulations, and guidelines of the CPC. In addition, a relatively specific management method has been formulated to control the readjustment of inefficient urban land use. However, the policies were mostly issued in the form of notices, opinions, and management methods, which have a low level of effectiveness.

**Table 4.** Purpose, process, and enforcement of policy-issuing institutions.

| Institutional Composition | Work Purpose | Work Process | Enforcement of Policies |
|---|---|---|---|
| CPC Central Committee | Implement major livelihood projects in full such as shantytown reconstruction, and promote urban renewal and equalization of urban-rural public services. | Central finance continued to increase its support for shantytown reconstruction and encouraged the National Development Bank and others to strengthen financial support. | Explored effective models for policy-based, developmental, and commercial financial institutions to support the readjustment of inefficient urban land use. |
| NPC of the People's Republic | Promote shantytown reconstruction in an orderly manner and improve supporting facilities. | Carry out surveys and evaluations of the intensive use of construction land and reduce the area of construction land used per unit of GDP. | The reconstruction of shantytowns was strengthened, and the goal of shantytown reconstruction affecting approximately 100 million people has been achieved. |
| State Council | Based on the second land survey, evaluate the utilization and the input-output situation of existing construction land and deal with existing problems in the utilization of construction land in accordance with laws and regulations. | Improve the supply method in the process of urban land readjustment; encourage former state-owned land users carry out renovation and readjustment. | If the land has been idle for more than one year and less than two years, the idle land fee will be levied at 20% of the land transfer price. |
| Ministry of Natural Resources | Carry out pilots for the readjustment of inefficient urban land use, revitalize low-efficiency urban land, increase the effective supply of urban construction land, and improve the ability of land to sustain economic and social development. | Identify 10 provinces including Inner Mongolia, Liaoning, Shanghai, Jiangsu, Zhejiang, Fujian, Jiangxi, Hubei, Sichuan, and Shaanxi to carry out pilot projects for readjustment of inefficient urban land use. | Improved incentive mechanism for the readjustment of inefficient urban land use, such as encouraging industrial transformation and upgrading to optimize the structure of land use. |
| Other Ministries | Encourage small-scale and gradual renewal of old urban areas to protect the city's traditional pattern and texture. | Comprehensive use of energy-saving transformation and other aspects of funds to accelerate the renovation of old communities. | Support the installation of elevators in old buildings that meet requirements to improve the building's use function and livability. |

A comparison of the horizontal and longitudinal composition of the policies shows that although the current policies for the readjustment of inefficient urban land use are not as clear as the policies for cultivated land protection and land use control in the Land Administration Law, a number of relevant policies have been jointly developed and promulgated by various departments such as the Ministry of Natural Resources. From an overall perspective, the frequency of joint promulgation reflects that the policies adopted by various management departments are based on system management considerations, and the coordination of relevant policies issued by various authorities will become the key to their smooth implementation.

### 3.4. Evolutionary Analysis of Policy Content

Based on the frequency of the promulgation of policies for the readjustment of inefficient urban land use in China (Table 2), the policy content presents an evolutionary trend from initial exploration to basic establishment and finally to continuous development, accompanied by the continuous enrichment of policy content and the increasing distribution of policies across the 10 categories (Figure 2).

### 3.4.1. Stage of Initial Exploration

Before 2013, the readjustment of inefficient urban land use was in the stage of initial exploration in China. The reform and open-up started in 1978, and China entered a development track characterized

by urbanization. Due to its focus on economic construction, the Chinese government ignored the shortcomings of rapid urban expansion and did not pay attention to the readjustment of inefficient urban land use. By the time of the promulgation and implementation of the Tentative Regulations of Urban Land Use Tax of the PRC in 1988, the Chinese government had gradually realized that urban construction land should follow the path of sustainable development. This regulation advocates the rational use of urban land, the adjustment of land-level income, the improvement of land use efficiency, and the strengthening of land management. In 2007, the Urban-Rural Planning Law of the PRC marks the first time that the Chinese government has clarified the principle of "reconstruction of the old city" in the form of law. In 2008, the Outline of the Master Plan for National Land Use (2006–2020) was issued. This master plan for land use was a top-level design to improve the readjustment of inefficient urban land use in China, and it marks entrance into a nascent stage. With the release of the 12th Five-Year Plan for National Economic and Social Development of the PRC (2011–2015) in 2011, the Chinese government was gradually turning its attention to the problem of economical and intensive land use. This laid a preliminary foundation for the readjustment of inefficient urban land use. The above policies did not directly address the readjustment of inefficient urban land use, but they clearly marked the beginning of the exploration of mechanisms for the readjustment of inefficient urban land use in China.

### 3.4.2. Stage of Basic Establishment

The period 2013–2015 was a stage characterized by the basic establishment of policy for the readjustment of inefficient urban land use in China. The promulgation and implementation of the Guiding Opinions on Piloting the Readjustment of Inefficient Urban Land Use in 2013 marked the formal establishment of the pilot project for the readjustment of inefficient urban land use in China. In 2014, the Guiding Opinions on Promoting the Relocation and Renovation of Old Industrial Areas in Urban Areas marked China's entrance into the climax stage of the "renovation of shantytowns". A series of issues such as extensive land use brought about by "shantytowns" drew great attention from the Chinese government [65,66]. In 2014, the promulgation of Guiding Opinions on Promoting Economical and Intensive Land Use signified that the Chinese government had standardized and guided the readjustment of inefficient urban land use at the institutional level. Subsequently, the Chinese government introduced a series of related policies until the promulgation of Recommendations of the CCCPC on the 13th Five-Year Plan for Economic and Social Development (2015), which marked the basic establishment of a policy system for the readjustment of inefficient urban land use in China. At this stage, the relevant policy system was initially constructed around the issues of Update and Renovation, Layout Optimization, Standard Control, and Market Configuration.

The two-dimensional map of the policy text (Figure 2) shows that the policies initially referring to only Layout Optimization, Revitalization and Utilization, Update and Renovation, Overall Planning and Standard Control evolve to include Update and Renovation, Layout Optimization, Standard Control, Market Configuration, Revitalization and Utilization, Overall Planning, Incentive Mechanism, Income Distribution, and Supervised Evaluation. In particular, the number of policies on the subject of market configuration increases significantly, accounting for 9.24% of the total. Moreover, themes such as Incentive Mechanism and Income Distribution continue to deepen, accounting for 6.9%, which indicates that the Chinese government is paying increasingly more attention to incentives and fairness in the readjustment of inefficient urban land use.

### 3.4.3. Stage of Continuous Improvement

In 2016–2019, the readjustment of inefficient urban land use in China was in the stage of continuous development. In 2016, under the Guiding Opinions on Deepening the Readjustment of Inefficient Urban Land Use (Trial), the Chinese government adhered to the principles of "government guidance, planning first", "market orientation, guided by the situation", and "public participation, equal consultation" [67]. This clarifies the scope of readjustment, promotes the investigation of inefficient urban land use and

the construction of the map, and further strengthens planning coordination and implementation. These guiding opinions mark the entry of the readjustment of inefficient urban land use into a new stage of development and improvement in China. In 2017, the Notice on Printing and Distributing the Promotion Plan for the Readjustment of Inefficient Urban Land Use (2017–2018) focused on establishing a mechanism for monitoring and evaluation, strengthening risk assessment, and promoting the readjustment of inefficient urban land use. Subsequently, the Key Points for the Preparation of Inefficient Urban Land Redevelopment Planning marked the first time a policy for the readjustment of inefficient urban land use planning was presented in the form of technical regulations in China.

In 2019, with the promulgation of the Several Opinions on Establishing a Territorial Planning System and Supervising Implementation, the Chinese government insisted on optimizing the spatial structure and layout of the territory, coordinating the comprehensive utilization of aboveground and underground space, and focusing on improving infrastructure and public service facilities, such as transportation and water conservancy. To promote the readjustment of inefficient urban land use under the new normal in China, the Ministry of Natural Resources issued the Notice on the Second Batch of Land-Saving Technologies and Model Case Collection, seeking typical cases of the readjustment of inefficient urban land use, stock construction and utilization, etc. This policy provides guidance for the development and improvement of the readjustment of inefficient urban land use in China.

The number of policy texts continues to increase, and the content is constantly enriched at this stage. In terms of the two-dimensional distribution of policy texts (Figure 2), 50% of the 10 total keywords are used, and the number of policy texts with the themes of Incentive Mechanism, Revitalization and Utilization and Update and Renovation has increased significantly since the previous stage, with increases of 8.79%, 8.79%, and 4.33%, respectively. Therefore, the readjustment of inefficient urban land use is clearly in the stage of continuous improvement in China. This means that under reasonable overall planning, the stock of construction land is revitalized, the incentive mechanism is optimized, and inefficiently used urban land is transformed and readjusted.

## 4. Conclusions and Discussion

In this study, our aim was to explore the evolution of policies concerning readjustment of inefficient urban land use in China and to provide a reference for adjustment and improvement of related policies. From the above results, the readjustment of inefficient urban land use has evolved from initial exploration to basic establishment and finally to continuous improvement. Although policies regarding readjustment of inefficient urban land use have gradually matured in China, opportunities for improvement and development remain in terms of Income Distribution, Supervised Evaluation, and Legal Liability. Considering the research questions proposed at the beginning of this article, we summarized the research results with respect to the theoretical framework, the evolution of policy contents, and theoretical contributions.

### 4.1. Changes in Policy Volume

Along with the rapid development of urbanization and industrialization, several relevant policies for readjustment of inefficient urban land use have been introduced to address the current situation of inefficient use of urban construction land in China. Throughout the evolution of relevant policies, although the Chinese government has actively implemented policies at the national level, many weaknesses remain compared with the urban renewal policies of European and American countries. Therefore, the Chinese government should further improve policies concerning readjustment of inefficient urban land use.

In terms of the time dimension, the number of relevant policies is increasing, from a total of 12 items between 1988 and 2012 to an annual average of 6 items between 2013 and 2019. Although the annual average number of relevant policies is increasing, a certain gap remains compared with the number of urban renewal policies in European and American countries, and many reasons may account for this gap. For example, judging from the social and urban development processes in China,

the United States, and Britain (Table 5), due to differences in the socio-economic environments in these countries, the development levels of related policies have differed. In the United States and Britain, the role of long-term market economic directly promotes the rapid development of cities. With the formulation and implementation of urban renewal policies, considerable changes have occurred in the urban environment in terms of the layout and spatial structure, and the problem of "urban disease" has been alleviated to varying degrees. However, throughout its 40 years of reform and expansion, China has experienced the entire modernization process carried out by Western developed countries. Urbanization is developing very quickly, but many problems related to inefficient land use remain. Accordingly, the Chinese government has continually promulgated policies for readjustment of inefficient urban land use to solve such problems. Although the urban spatial layout has improved, a considerable amount of work is still required [68,69].

**Table 5.** Comparison of relevant working environments in China, the United States, and Britain.

| Conditions | China | the United States | Britain |
|---|---|---|---|
| Land ownership | Socialist public land ownership shapes state land ownership and collective land ownership. | Private land ownership predominates, with only a small amount of state-owned land. | British land is nominally owned by the British royal family. At present, the main form of land ownership in the United Kingdom is socage or freehold [70]. |
| Environmental evaluation process | According to compiled units of regulatory planning, the scope of environmental assessments is determined, and partial assessments can be carried out for sporadic projects targeting readjustment of inefficient urban land use. | The first phase of environmental assessment is a survey of the brownfield history and a description of the current situation. Pollution sites and types determined by the second stage of environmental assessment will have important impacts on end use and land development planning [71]. | In addition to official assessments and summaries conducted by the British government, other general assessments of urban renewal and redevelopment are carried out. |
| Financial resources | Government funds are the main form of support, and industrial funds and asset-backed securities (ABS) provide financing for the initiation of related projects. | The nobility of the private market plays a key role in reviving urban centers. | The government created urban renewal companies to attract investments through collaboration between the private and public sectors. |
| Social participation mechanism | In addition to the government's leading role, people/entities with state-owned land use rights, collective economic organizations and other market entities are encouraged to participate in urban development. | The government and developers form a "Smart Growth" partnership. | The participation of local communities and cooperation among public, private, and community entities are emphasized. |
| Urbanization process | China's urbanization process conforms to low-level urbanization. This means the level of urbanization lags behind the level of economic development [72]. | Urbanization of the United States has entered the suburbanization stage. The suburbanization of the United States reflects natural expansion from the core of the city. The driving forces are commercial interests and consumer preferences. | Urbanization in the UK has entered an orderly development stage, changing from spontaneity to planning. The concept of urbanization in Britain has changed from the original city-centric model to a people-centered model. |

## 4.2. Longitudinal and Horizontal Policy Structures

In terms of the policy-promulgating agencies, the number of policies promulgated by the CPC Central Committee, the NPC of the People's Republic, the State Council, and relevant ministries account for 13.46%, 11.54%, 28.85%, and 46.15% of all policies, respectively. Overall, the CPC Central Committee and the NPC of the People's Republic have played vital roles in formulating policies for readjustment of inefficient urban land use in China. In response to the call of the CPC Central Committee and the NPC of the People's Republic, the State Council, the Ministry of Natural Resources, and other ministries have actively engaged in readjustment of inefficient urban land use. Because the work is mainly government-oriented, the degree of social participation is not very high compared with that in European and American countries. Urban renewal in America and Britain (Table 5), the idea of cooperation and participation among the government, private sectors and the community have been widely accepted by residents. A long-term model of urban renewal characterized by cooperation among

the government, private sectors, and the community has been established, which has promoted the virtuous cycle of urban renewal in a market-oriented environment. However, in China, readjustment of inefficient urban land use is driven by the government. Relevant laws and regulations are imperfect, and social participation is insufficient. Related work still focuses on the pursuit of economic returns and lacks a comprehensive and integrated concept of urban development.

In terms of the policy text form, the existing policies are mostly issued in the form of Notices and Opinions. The number of relevant policies issued in the form of Guidelines of the CPC central committee, Laws, and Regulations is relatively small. This phenomenon is mainly due to the hierarchical system of Chinese government agencies. The CPC Central Committee issues guiding documents, the NPC of the People's Republic formulates and revises relevant laws, the State Council formulates and revises administrative regulations and manages various ministries according to relevant laws, and the Ministry of Natural Resources and other ministries are agencies of policy refinement and enforcement. It is not difficult to see that with the higher level of institutions, the policies number of the readjustment of inefficient urban land use issued by relevant institutions is getting smaller. Therefore, the level of effectiveness of related work is low, and the normative role is not strong.

The current system of urban management in China is flawed and lacks reasonable institutional policies for support and guidance. Too many management agencies have seriously affected communication between the government and enterprises, which has impacted the progress of the readjustment of inefficient urban land use in China. Therefore, the Chinese government should optimize the design of the top-level system and accelerate system legislation for relevant work, especially system legislation for local governments, to strengthen and regulate policy guidance among local governments.

### 4.3. Evolution of Related Policies and Contents

The readjustment of inefficient urban land use has evolved from initial exploration to basic establishment and finally to continuous improvement in China. On the whole, the policy content has basically been improved, with themes such as Standard Control, Market Configuration, Overall Planning, Incentive Mechanisms, Income Distribution, and Supervised Evaluation. According to the analysis of the above results, the Chinese government has increasingly focused on issues related to the Incentive Mechanisms and Market Configuration of related work. Such work addresses inefficient urban land use readjustment mainly through land rent, price leverage, taxation, planning adjustments, land reserves, and other methods of market allocation [73]. Although policies regarding readjustment of inefficient urban land use have gradually matured in China, opportunities for exploration and development remain in terms of Income Distribution, Supervised Evaluation, and Legal Liability. Accordingly, the Chinese government should promulgate as many policy texts oriented toward specific aspects of implementation as possible, especially from the perspectives of Income Distribution, Supervised Evaluation, and Legal Liability, such as technical regulations, to strengthen measures for policy implementation and to ensure smooth progress in the readjustment of inefficient urban land use. In addition, in terms of management agencies, this paper describes the evolution from centralized management of the Ministry of Natural Resources to cooperative management of this ministry by several entities, reflecting the joint responsibility of multiple sectors for the readjustment of inefficient urban land use in China [74–76].

Existing policies mainly target state-owned construction land, and a standard system integrating Standard Control, Market Configuration, and Incentive Mechanisms has been formed. In the context of current territorial planning in China, to create new value and revitalize space, attention will be directed toward inefficiently used land and stock land across the entire territorial space.

### 4.4. Theoretical Contributions, Method Limitations, and Directions of Further Research

This paper fundamentally contributes to systematic analyses of the evolution of policies concerning the readjustment of inefficient urban land use in China and uses a research method combining

quantitative and qualitative approaches. This paper used the content analysis method to build a theoretical analysis framework based on promulgation dates, the longitudinal hierarchy and the horizontal compositions of policies, identify analytical categories, and analyze the status quo and evolution of policy contents based on quantitative analysis, thus overcoming the subjectivity and uncertainty of qualitative research to a certain extent. Combined with NVivo 12 qualitative analysis software, we obtained empirical conclusions through statistical analysis of word frequencies, social network analysis and other analytical methods, compared and verified the results obtained by the content analysis method, and provided new insight into current policies through systematic analysis of the evolution of policies concerning the readjustment of inefficient urban land use in China.

This paper mainly uses the content analysis method to analyze the evolution of policy concerning the readjustment of inefficient urban land use in China. Although this method can transform qualitative analysis into quantitative research, there are still some limitations. For example, during the formulation of the theoretical framework, the entire development process of the policy can be divided into multiple stages based on the promulgation date. Although the division of the time period is mainly determined by combining the knowledge of relevant experts, this process is somewhat subjective and uncertain.

Further research directions are suggested to improve analysis of the effectiveness of the policy, to resolve the limitations of the content analysis method, and to further improve the ability to carry out quantitative research in this field to effectively assess the policies for the readjustment of inefficient urban land use.

**Author Contributions:** Y.B. and Y.G. conceived of and designed the research; Y.B. and X.L. made substantial contributions to the design, data processing, and analysis; W.Z. and W.H. critically revised the paper; Y.B. and B.H. contributed to data processing and analysis; F.L. and H.Y. contributed the original data; Y.B. wrote the paper; and W.Z. gave valuable suggestions for the revision of the paper. All authors read and approved the final manuscript.

**Funding:** This research received no external funding.

**Acknowledgments:** We sincerely thank the Land Consolidation and Rehabilitation Center of the Ministry of Natural Resources for supplying the research data and the anonymous reviewers and editors for offering valuable comments.

**Conflicts of Interest:** The authors declare no conflicts of interest.

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
