# Peer review of "Evolution of Policy Concerning the Readjustment of Inefficient Urban Land Use in China Based on a Content Analysis Method"

_sustainability, doi:10.3390/su12030797_

Round 1
Reviewer 1 Report
This paper reviews the Chinese policies for readjusting inefficient urban land uses. It is an interesting topic to examine. But in general I suggest to address more on the evaluation of policies on different levels and time periods, and discuss the essential reasons for the results. English and style editing are needed.
I have the following comments:
It needs a brief introduction of the research background and research outcomes significance in the abstract. Line 43, please give more explanation of planning shifting from incremental expansion to stock exploration. (Does it mean the shift from urban sprawl to urban land redevelopment?) Line 46, please add "and" before starts Line 47, please add "infrastructure" after upgrading. Line 51, please remove "," Line 51. Intensive, scientific, balanced what? (development modes?) Line 54. It needs more information or data to support the readjustment of inefficient urban land uses is an inevitable choice (for example what is the average percentage of inefficient urban land in China currently). The fundamental differences between the Western urban renewal process and the new stage of readjusting inefficient urban land uses in Chinese coastal areas need to be addressed to support the claim from Line 60-69. Line 70, please replace "a" with "into" The research question needs to be addressed in the Introduction. Line 148-151, what is the standards or rationality to divide into these 10 analytical categories. Or what is the relationship of these 10 categories with the inefficient land use readjustment process in China. The brief summary of of the 52 policies need to be listed in Table 1 and discussed in the results and analysis session to enhance this review's depth and novelty。 Line 174. Please ass more information to explain the experts background and selection reasons. The reform and open-up reform starts from 1978. Please explain the rationality to discuss in the time period 1988-2012, 2013-2015, 2016-2019. What is the reason for the number difference in different time period. Line 193-200. It needs to add the comparison and contrast of the different purpose, process and enforcement of policy-issuing institutions. Just the number difference of policies on different levels is weak. Line 211-218. Please provide details and facts to demonstrate that the readjustment policy is not as clear as land use control. Line 226-245. Please add reference of the citation. The discussion session is just stating the content of policies issued in different part. I suggest to move this part to the content description in the previous session and focus on discussing the evaluation of the policies and the reasons for the policies changes and evolution. Line 341-342, 355-357,Please add reference to the citation. The suggestions made in the conclusion (e.g. Line 349-350; 353-357; 361-367) regarding the future policies could not be clearly deduced from the analysis.Author Response
Please see the attachment.

Reviewer 2 Report
Particularly noteworthy is the review of the literature and the selection of data sources. The method of analysis and presentation of test results should also be highly rated.
However, there are a few minor shortcomings in the work. First of all, the introductory part did not explicitly specify the purpose of the study - it was only outlined in the abstract. The summary of the introductory part of the article indicates the next stages of work, but here it should be more articulated the research tasks associated with each of the subsequent stages of work. Due to the review nature of the study, authors should also further highlight the innovation of the study.
In the methodological and resultant part, the selection and differentiation of the length of time periods is unclear. Why was the 2013-2019 split into two unequal periods? The name of the "Before 2012" interval suggests that it has been included in the data ending in 2011. Therefore, which of the ranges does the data for 2012 fall in? Perhaps it is a simplification or an error in separating or naming the range. The uncertainties associated with time periods should be clarified in the chapter describing the methods.
In addition, the last part of the article should be developed. The discussion should refer to the current research to a greater extent, but apart from conclusions, this part of the work should also be summarized.
Reviewer 3 Report
The paper seems to me very interesting, also because it extended to a formidable territorial dimension like the Chinese, incredibly rich in problems and criticalities. From the examination of the text, however, some clarification passages should be made which in my opinion are the following.
A first observation that emerges is that the introduction completely lacks any data that gives the idea of the quantities involved. it is clear that the objective of the work is not this of discussing quantities, but in the Chinese case it is essential to report the orders of magnitude of urbanization that have no equal in the world.
From this point of view I believe it is also necessary to review some statements in paragraph 2.1. Content Analysis Method
In table 1 I believe it is appropriate to insert the years of the provisions in a separate column, so as to be able to better visualize how they densify or concentrate in some periods compared to others.
It would also be very interesting if these listed measures could also provide a qualitative judgment of efficiency, possibly formulated in an expert-based manner by the authors.
Figure 1 is not very clear as the correspondences between the elements represented on the three axes cannot be directly interpreted.
In Table 2 it would be advisable to change the X and Y codes with others to mean frequency and percentage, since, normally, x and y are used to name unknown variables.
Fig. 2 is definitely complicated to interpret. perhaps the clusters that appear here would be better to reorganize them directly in the table where the numbering of the measures are attributed (Table 1)
Also Fig. 4 appears to be chaotic and not very informative. perhaps it would be better to include the indications it contains in two separate graphs.
Reviewer 4 Report
Dear Authors, I found the article interesting and novel, although focused on the Chinese case. The paper should be improved before acceptance. Take in mind that Sustainability is an international journal and every case study should be interesting also for readers in other continents. First, I see that your article should be classified as case report, not as article as it is now. Second, in my opinion the policy effectiveness analysis is in some way underdeveloped since it is mainly qualitative. I agree there are difficulties to quantify effectiveness of policy interventions on the field (e.g. land consolidation), but I would see at least a honest discussion about the possibility to develop further quantitative research on this field in order to effectively assess the ability of a given policy to improve land-use efficiency. Third, I would see an enriched definition of land-use efficiency. I don't see any clear and definite notion about this concept in the present version. Fourth, I would see an enriched introduction and discussion with a wide review (state of the art) of the international situation about land policies. More relevant examples should be selected and presented in order to allow readers to evaluate the representativeness (or not) of the Chinese case.
And last but not least, a literature review needs also some improvements in the literature list, considering less grey literature and increasing the number of seminal articles/books cited. Language usage should be also checked extensively.
Round 2
Reviewer 1 Report
The paper gets improved after the revision.
But I think the research question raised in Line 118-122 is not answered or discussed in the paper yet. If the research is "how", I would suggest to compare and apply different literature review and statistic methods to generate new insight into the current policy in China. Because the paper selects the content analysis method directly, I think the statement is not convincing that the paper systematically analyze the relevant policies so far.
The results is not discussed sufficiently. I suggest to compare the Chinese policies review results with other countries' policies to generate novel insights.
Overall I am still very confusing is it a research article or literature review. The title is more like an article, but it lacks of a clear research question and appropriate methods. I suggest to change it to a review paper.
Reviewer 3 Report
I think that the authors have inserted the changes according to my revision indications. As far as I am concerned the work can be published in the present form.
Reviewer 4 Report
Good revision. Pay attention to language correctedness and style formats.
Round 3
Reviewer 1 Report
Dear Authors,
Thank you for resubmitting the paper.
I think the paper has included enough information to introduce and describe the land use policies in China. But I still have two major concerns.
First, the research questions from Line 119-121 are not explored thoroughly. If the research question is "how". It need at least to compare different methods or ways to do the literature review, evaluate the different methods in discussion, and conclude with the answer to "how", rather than making the decision steps to follow from Line 124-132.
Second, the comparison of Chinese policy with other countries's policy in discussion needs evidence support. Using the "west" as representative does not make sense. I suggest to compare with some countries explicitly due to the conditions of different land ownership, environmental evaluation process, funds resource. etc.
Thank you
Round 4
Reviewer 1 Report
It is improved